environmental science/biochemistry/
environmental chemistry

*Acinetobacter johnsonii*, partial denitrification,
double-substrate inhibition model, real-time
polymerase chain reaction

**Author for correspondence:**
Jun Li
e-mail: bjut_lijun@163.com

This article has been edited by the Royal Society of Chemistry, including the commissioning, peer review process and editorial aspects up to the point of acceptance.

# Investigation of growth kinetics and partial denitrification performance in strain *Acinetobacter johnsonii* under different environmental conditions

Yang Zhang, Xiujie Wang, Weiqi Wang, Zhitao Sun and Jun Li

The College of Architecture and Civil Engineering, Beijing University of Technology, Beijing 100124, People's Republic of China

(iD) YZ, 0000-0002-2139-5860; JL, 0000-0001-9470-6140

A denitrifying strain ZY04 with a high nitrite-accumulating rate was isolated and purified from activated sludge in a laboratory-scale $A^2/O$ reactor. The strain was characterized and identified as *Acinetobacter johnsonii* by 16S rDNA phylogenetic analysis. The sequences of the key functional genes (*napA*, *nirB*, *nirD*) involved in partial denitrification were amplified via polymerase chain reaction, which provided a basis for exploring gene expression. The effects of different environmental factors (C/N ratio, pH and temperature) on the partial denitrification performance and transcriptional levels of the functional genes during the logarithmic growth phase were investigated by batch experiments. The results showed that the partial denitrification performance was optimal when the C/N ratio was 5, the pH value was 6–8 and the temperature was 25°C. The gene expression during the logarithmic growth phase indicated the good performance of partial denitrification under different environmental conditions. All three functional genes exhibited the highest expression levels at 25°C. The results of inhibitory kinetics analysis revealed that three biokinetic models (Aiba, Edwards and Andrews) simulated the growth pattern of strain ZY04 inhibited by a single substrate (nitrate or sodium acetate) well. In the double-substrate inhibitory model, five models of nine combinations successfully fitted the growth characteristics of the strain affected by the double substrate of nitrate and sodium acetate. The relevant semi-saturation parameters and substrate inhibition parameters were obtained, and the correlation coefficient ($R^2$) reached 98%.

# 1. Introduction

ANAMMOX (anaerobic ammonium oxidation) is a novel nitrogen removal technique that was proposed by Mulder *et al.* [1] in 1995. The technique has the advantages of low energy consumption, low sludge output and high nitrogen removal efficiency [2,3]. This process uses nitrite ($NO_2^- $-N) as the electron acceptor to oxidize ammonia ($NH_4^+$-N) into nitrogen ($N_2$) or uses $NH_4^+$-N as the electron donor to reduce $NO_2^-$-N into $N_2$ [1,4]. In biological nitrogen removal processes, nitrite can be obtained in two ways, namely by short-cut nitrification ($NH_4^+ \rightarrow NO_2^-$) or by partial denitrification ($NO_3^- \rightarrow NO_2^-$). Nitrite, an important product in short-cut nitrification, is usually accumulated by inhibiting nitrite-oxidizing bacteria and conserving ammonia-oxidizing bacteria. However, further oxidation of nitrite is inevitable, and autotrophic nitrifying bacteria in the short-cut nitrification process are sensitive to the environment. These disadvantages result in difficulty in maintaining the stable operation of short-cut nitrification. The denitrification process reduces $NO_3^-$-N into $NO_2^-$-N, catalysed by nitrate reductase, and further reduces $NO_2^-$-N into gaseous nitrogen (NO, $N_2O$ or $N_2$), catalysed by nitrite reductase. Therefore, the accumulation of nitrite can be realized by controlling the process of $NO_3^- \rightarrow NO_2^-$ to be dominant, which is defined as partial denitrification, to provide the necessary substrate for ANAMMOX. The heterotrophic denitrifying bacteria in the partial denitrification process have a short generation cycle and a significant impact on load resistance. Additionally, the partial denitrification process can consume organic carbon sources and create appropriate conditions for autotrophic ANAMMOX by reducing the inhibitory effect of organic matter. The provision of a nitrite substrate for ANAMMOX by partial denitrification is considered to be the most promising and energy-saving process for wastewater treatment [5–7]; and it also has great significance for the treatment of chemical fertilizer wastewater, explosives wastewater and other production wastewater [8,9]. Theoretically, 1.41 mg chemical oxygen demand is needed to remove 1 mg nitrite, which could reduce the facility cost by more than half [10]. Previous works have indicated the ultra-high nitrite removal rate of partial denitrification [7] and a study by Du *et al.* [11] proved that the nitrogen removal efficiency of the partial denitrification-ANAMMOX process driven by acetate could reach 93.6% in the treatment of synthetic wastewater. Moreover, the negative influence of high-strength $NO_3^-$-N and $NO_2^-$-N on denitrifying bacteria could be alleviated by combining partial denitrification with ANAMMOX [12]. Cao *et al.* [13] studied the granulation of partial denitrifying sludge to reduce the volume of the denitrification tank. Cao *et al.* [14] successfully cultivated denitrifying sludge with high nitrite accumulation in the laboratory; the nitrate transformation rate reached up to 80% and maintained a long-term operation of 300 days. This provides a foundation for the treatment of nitrate wastewater by ANAMMOX. Wang *et al.* [15] achieved the accumulation of nitrite in an activated sludge system by controlling the C/N ratio, sludge retention time and carbon source type. Several reported studies [16–18] have realized the synergistic effect of denitrification and ANAMMOX in laboratory-scale reactors. The substrate inhibitory model of denitrifying strains using nitrate and carbon sources as substrates has been discussed in previous literature [9,19], but single-substrate dynamics, unlike double-substrate dynamics, may ignore the interactions between the two substrates [20–22] and the established model may not accurately reflect the microbial characteristics. Therefore, it is necessary to establish a double-substrate model to describe the relationship between the substrate concentrations and the growth characteristics of microorganisms.

This paper explores the influence of metabolic substrates (carbon or nitrogen source) on denitrifying bacteria, the partial denitrification performance and differences in functional gene expression under different environmental conditions from the pure-cultured bacteria point of view. In the heterotrophic denitrification process, the availability of the carbon source in terms of the C/N ratio has been found to influence both the nitrate reduction pathway and the carbon utilization patterns, resulting in different nitrate removal efficiencies [23,24]. Glass & Silverstein [9] observed that denitrification was significantly inhibited at pH 6.5 and that the nucleophilic addition reaction increased significantly as the pH increased from 7.5 to 9.0. Qian *et al.* [25] considered that a high pH may benefit the accumulation of nitrite during denitrification owing to alkalization. The effect of temperature on denitrification was found to be greater than that for other biological wastewater treatment processes; the appropriate temperature range was 15–35°C, and the denitrification rate decreased significantly when the temperature was lower than 10°C [26]. For these reasons, the C/N ratio, pH value and temperature were considered as the environmental factors in this experiment. A denitrifying strain *Acinetobacter johnsonii* ZY04 with a high nitrite-accumulating rate was isolated and purified from activated sludge in a laboratory-scale $A^2$/O reactor. Three different inhibitory models of the substrate (Aiba, Edwards and Andrews models) were used to simulate the effects of necessary substrates (nitrate and carbon sources) on bacterial growth. A single-substrate inhibitory kinetic model and a double-substrate kinetic model were, respectively, established to obtain the relevant kinetic parameters. The effects of different C/N ratios, initial pH values and

temperatures on denitrification performance, partial denitrification abilities and gene expression levels of the strain were studied by batch experiments. It is hoped that this study can provide a theoretical basis and new ideas for the implementation of partial denitrification and its coupling with ANAMMOX.

# 2. Material and methods

## 2.1. Culture media

The Luria–Bertani (LB) medium was composed of $10.0 \text{ g l}^{-1}$ tryptone, $5.0 \text{ g l}^{-1}$ yeast extract and $10.0 \text{ g l}^{-1}$ NaCl and had a pH of 7.0–7.2. The denitrification medium was composed of $1.2 \text{ g l}^{-1}$ $CH_3COOH$, $0.7 \text{ g l}^{-1}$ $KNO_3$, $0.6 \text{ g l}^{-1}$ $K_2HPO_4$, $0.2 \text{ g l}^{-1}$ $MgSO_4 \cdot 7H_2O$ and 2.5 ml trace elements solution and had a pH of 7.0–7.2. The trace element solution was composed of $50.0 \text{ g l}^{-1}$ $Na_2EDTA$, $2.2 \text{ g l}^{-1}$ $ZnSO_4 \cdot 7H_2O$, $5.5 \text{ g l}^{-1}$ $CaCl_2$, $5.0 \text{ g l}^{-1}$ $MnCl \cdot 4H_2O$, $5.0 \text{ g l}^{-1}$ $FeSO_4$, $1.6 \text{ g l}^{-1}$ $CuSO_4 \cdot 5H_2O$ and $1.6 \text{ g l}^{-1}$ $CoCl_2 \cdot 6H_2O$. Agar 1.5–2% was added to the solid medium, and all the media were sterilized at 120°C for 15 min before use.

## 2.2. Isolation and identification of strain

Activated sludge taken from a laboratory-scale $A^2/O$ reactor was added to a sterilized 0.9% NaCl solution, and the mixed suspension was coated on LB agar plates by the gradient method. After culturing for 48–60 h at 30°C, the single colonies on the plates with strong growth and different morphological appearances were numbered and selected for further culturing. The selected colonies were cultivated in the liquid denitrification medium, and samples were taken periodically to detect the contents of nitrate, nitrite and total organic carbon (TOC). The strain ZY04 with a stable and high nitrite-accumulating rate during the denitrification process was selected for subsequent study and stored in 30% glycerol solution at −80°C.

The morphology of the strain was observed on a solid plate, and Gram staining of the cells was carried out. To identify the strain by the 16S rDNA gene, a single colony was chosen and cultured in liquid denitrification medium for 24 h at 30°C while maintaining the shaking speed of the incubator at 160 rpm. Next, 1 ml bacterial solution was placed into a 1.5 ml centrifuge tube and centrifuged at 8000 rpm to collect the sediment. The genomic DNA of strain ZY04 was extracted using an Ezup Column Bacterial Genomic DNA Extraction Kit (Sangon Biotech, Shanghai, China), and the purified DNA was used to amplify the 16S rRNA gene by polymerase chain reaction (PCR) on a Thermal Cycler (Bio-Rad iCycler, USA) using bacterial universal primers, as shown in table 1. The PCR products were sent to Sangon Biotech (Shanghai, China) for sequencing, and the BLAST online program on the NCBI website (https://www.ncbi.nlm.nih.gov/) was used to compare the resulting sequences with the sequences of existing strains in GenBank. Sequences with high homology were selected and used to construct a phylogeny tree with the sequence of ZY04 via MEGA 5.0 software.

## 2.3. Denitrification characteristics under different environmental conditions

### 2.3.1. Effect of C/N ratio on denitrification performance

The C/N ratios were set as 3, 4, 5, 6 and 7 for experimentation. The sodium acetate concentrations were changed under the condition that the initial nitrate concentrations were fixed at $100 \text{ mg l}^{-1}$. The pH value was adjusted to about 7.5. The overnight cultured suspension ($OD_{600} \approx 0.5$) was inoculated into a serum bottle at a 5% inoculation rate ($v/v$) and cultured at 30°C in an incubator with a shaking speeding of 150 rpm. The media without the inoculum were set as control groups, and every experiment was conducted in triplicate. The $OD_{600}$, nitrate, nitrite and TOC were determined regularly after centrifugation.

### 2.3.2. Effect of pH value on denitrification performance

The pH values were set as 6, 7, 8, 9 and 10 for experimentation. The C/N ratio was set as 5 with an initial nitrate concentration of $100 \text{ mg l}^{-1}$. The overnight cultured suspension ($OD_{600} \approx 0.5$) was inoculated at a 5% inoculation rate ($v/v$) and cultured at 30°C with a shaking speed of 150 rpm. The media without the inoculum were set as control groups, and every experiment was conducted in triplicate. The $OD_{600}$, nitrate, nitrite and TOC were determined regularly after centrifugation.

**Table 1.** PCR primer list.

| gene name | primer sequence (5'–3') | product size (bp) |
|---|---|---|
| *16S* [27,28] | AGAGTTTGATCCTGGCTCAG | 1441 |
| | TACGGYTACCTTGTTACGACTT | |
| *napA* | ACGGCACGGTGACTAATTCTGAAC | 290 |
| | CGTCGGTTCAAGCTCTTGGTAGTC | |
| *nirD* | GTGGTAATCGGTGGTGGACTGTTG | 300 |
| | AAGTGCCATGTTCGGACGAATACC | |
| *nirB* | GGCAGTACATGGTGTCGCTACG | 206 |
| | CCGCCATTACCGCAGACATACAG | |
| *16S*[a] | CAGCMGCCGCGGTAATWC | 468 |
| | CCGTCAATTCMTTTRAGTTT | |
| *napA*[a] | AGAACGCCCAGCCCAAGC | 45 |
| | TGTGAAACTGACCACCAATCC | |
| *nirD*[a] | GGTGGACTGTTGGGACTTG | 121 |
| | GCATTTGGCTGGCTTTACT | |
| *nirB*[a] | GGATGACTCGGTGGGTTTA | 105 |
| | GCACATTCACGGGTACAGC | |

[a]The primers were used in real-time PCR.

### 2.3.3. Effect of temperature on denitrification performance

The temperatures were set as 20°C, 25°C, 30°C, 35°C and 40°C for experimentation. The C/N ratio was set as 5 with an initial nitrate concentration of 100 mg l$^{-1}$. The pH value was adjusted to about 7.5. The inoculum (OD$_{600} \approx 0.5$) was inoculated into a serum bottle at a 5% inoculation rate ($v/v$) and cultivated in an incubator with a shaking speed of 150 rpm. The media without the inoculum were set as control groups, and every experiment was conducted in triplicate. The OD$_{600}$, nitrate, nitrite and TOC were determined regularly after centrifugation. All media were sterilized before use.

### 2.3.4. Calculation formulae

Nitrite accumulation efficiency:

$$a = \frac{C_{1i} - C_{10}}{C_{20} - C_{2i}} \times 100. \tag{2.1}$$

Nitrate removal efficiency:

$$b = \frac{C_{20} - C_{2i}}{C_{20}} \times 100. \tag{2.2}$$

Specific nitrate degradation rate:

$$c = \frac{C_{20} - C_{2i}}{\Delta t} \times \frac{1}{CC_0}. \tag{2.3}$$

Specific nitrate degradation rate:

$$d = \frac{C_{20} - C_{2i} + C_{10} - C_{1i}}{\Delta t} \times \frac{1}{CC_0}. \tag{2.4}$$

In the above equations, $a$ is the nitrite accumulation rate (%), $b$ is the nitrate removal rate (%), $c$ is the specific nitrate degradation rate (g (g DCW h)$^{-1}$), $d$ is the specific nitrate degradation rate (g (g DCW h)$^{-1}$), $C_{1i}$ is the nitrite concentration at a given time (mg l$^{-1}$), $C_{10}$ is the initial concentration of nitrite (mg l$^{-1}$), $C_{2i}$ is the nitrate concentration at a given time (mg l$^{-1}$), $C_{20}$ is the initial concentration of nitrate nitrogen (mg l$^{-1}$), $\Delta t$ is the time from the initial time to a given time (h) and $CC_0$ is the initial cell concentration (mg l$^{-1}$).

## 2.4. Amplification of functional genes and detection of transcriptional levels

The DNA was extracted as a PCR template using the method outlined in §3.2. The functional gene fragments were predicted based on the entire genome sequence of strain ZY04 (Y. Zhang 2018, unpublished data), and the primers were designed using the online primer design tool of Sangon Biotech (Shanghai, China). The primer sequences are presented in table 1. The PCR protocol of the nitrate reductase gene consisted of the following steps: 94°C for 5 min, 35 cycles at 94°C for 30 s, annealing temperature of 50°C for 40 s, 72°C for 90 s and final extension of 72°C for 7 min. The PCR protocol of the nitrate reductase gene consisted of the following steps: 95°C for 6 min, 10 cycles at 95°C for 30 s, annealing temperature of 60°C for 30 s (the annealing temperature was decreased by 0.5°C after every cycle), 72°C for 30 s, 25 cycles at 90°C for 30 s, annealing temperature of 55°C for 30 s, 72°C for 30 s and final extension of 72°C for 7 min.

The transcriptional differences of the functional genes during the denitrification process were detected using the real-time PCR method based on the SYBR Green I system. Samples were taken under different cultured conditions at 10 h (during the logarithmic growth phase) and centrifuged at 8000 rpm for 10 min. The RNA was extracted using a bacterial RNA extraction kit (Tiangen, Beijing, China). The PrimeScript RT Reagent Kit with gDNA Eraser was used for cDNA reverse transcription according to the manufacturer's instructions. The real-time PCR reaction system was as follows: 10 µl 2 × Master Mix, 0.5 µl forward primer (10 µM), 0.5 µl reverse primer (10 µM), add ddH$_2$O up to 18 µl. After mixing and centrifuging, 2 µl cDNA was added into the system. The mixture was put into a 96-well PCR plate on the real-time PCR instrument. The PCR reaction was performed according to the following procedure: 95°C for 30 s, 40 cycles at 95°C for 5 s and 60°C for 40 s for fluorescent collection; 95°C for 10 s, 60°C for 60 s, 95°C for 15 s and slowly heating from 60°C to 99°C to establish the melting curve. All samples were tested in triplicate, and the results were analysed using the $2^{-\triangle\triangle\text{ct}}$ method. The samples cultured under the conditions of a C/N ratio of 5, pH value of 8 and temperature of 30°C were set as the control groups.

## 2.5. Modelling of microbial growth kinetics

Kinetic analysis, especially the model simulation of substrate impact, is commonly used in the Monod equation [29–31] and its derived Teissier [21], Moser [32] and Contois [29] equations. In the present study, the dynamic analysis of substrate inhibition for cell growth was conducted by setting a larger scale of initial concentrations than have been set previously. The strain was cultured in 150 ml liquid denitrification medium with an inoculation amount of 5% (v/v) at 30°C with a shaking speed of 160 rpm. The C/N ratio in this experiment was set as 5, and the concentrations of the nitrogen and carbon sources were changed gradually. Samples were taken regularly for determination. The initial nitrate concentrations were set as 50, 100, 150, 200, 250, 300, 400, 500, 750, 1000, 1500 or 2000 mg l$^{-1}$, respectively, and the carbon source concentrations were 250, 500, 750, 1000, 1250, 1500, 2000, 2500, 3750, 5000, 7500 or 10 000 mg l$^{-1}$, accordingly.

The specific growth rate is calculated by

$$\mu = \frac{\ln X_2 - \ln X_1}{t_2 - t_1}. \tag{2.5}$$

In the above equation, $\mu$ is the specific growth rate of the strain (h$^{-1}$), $X_1$ is the dry cell weight of the strain at time $t_1$ (mg l$^{-1}$), $X_2$ is the dry cell weight of the strain at time $t_2$ (mg l$^{-1}$), $t_2$ is the initial time of any stage of strain cultivation (h) and $t_1$ is the end time of any stage of strain cultivation (h).

Three kinds of substrate inhibitory models, namely Andrews [33], Aiba [34] and Edwards [35], were used to fit the experimental data. To model single-substrate limiting growth kinetics based on NO$_3^-$-N or TOC, the following three growth kinetics equations (equations (2.6)–(2.8)) were, respectively, used:

$$\text{Andrews:} \quad \mu = \frac{\mu_{\max}}{1 + (K_s/S) + (S/K_i)}, \tag{2.6}$$

$$\text{Aiba:} \quad \mu = \mu_{\max}S \cdot \frac{\exp(-S/K_i)}{K_s + S} \tag{2.7}$$

and

$$\text{Edwards:} \quad \mu = \mu_{\max}\left[\exp\left(\frac{-S}{K_i}\right) - \exp\left(\frac{-S}{K_s}\right)\right], \tag{2.8}$$

where $\mu$ is the specific growth rate (h$^{-1}$) of the strain, $\mu_{\max}$ is the maximum specific growth rate (h$^{-1}$) of the strain, $S$ is the initial concentration of the substrate (mg l$^{-1}$), $K_s$ is the semi-saturation constant of the substrate and $K_i$ is the substrate inhibition constant.

Double-substrate limiting growth kinetic models were developed to examine whether NO$_3^-$-N and TOC were both limiting substrates. Therefore, every two kinetic equations from equations (2.6)–(2.8)

that were used to describe $NO_3^-$-N or TOC were inserted into the interactive form of the multiple-substrate growth models (equation 2.9) as follows:

$$\frac{\mu}{\mu_{\max}} = [\mu(S_1)][\mu(S_2)] \cdots [\mu(S_i)]. \tag{2.9}$$

According to the growth conditions of strains at different temperatures, the modified Arrhenius equation [36] was used to describe the influence of temperature on the growth rates of strains,

$$\mu_{\max} = A e^{-E/RT}, \tag{2.10}$$

where $\mu_{\max}$ is the maximum specific growth rate ($h^{-1}$) of the strain, $A$ is a pre-exponential factor ($days^{-1}$), $E$ is the activation energy ($kJ \, mol^{-1}$), $R$ is the gas constant 8.31 ($kJ \, (kmol)^{-1}$) and $T$ is the absolute temperature (K).

## 2.6. Analytical methods

The concentrations of $NO_2^-$-N and $NO_3^-$-N were analysed in accordance with the standard methods [37]. The cell density was described by the absorbance of the bacterial suspension at a wavelength of 600 nm ($OD_{600}$). The dry cell weight was determined by drying the cultivated cells at 105°C after washing and centrifugation. The temperature and pH value were detected using a WTW/Multi 3420 multiparameter device. Figures and dynamic model fitting were constructed using Excel 2016 and Origin 2018 software. All data were expressed as the mean ± s.d. One-way ANOVA was carried out in SPSS software for statistical analysis. The differences between samples were considered significant at $p < 0.05$.

# 3. Results and discussion

## 3.1. Isolation and identification of ZY04

A total of 36 strains with different morphologies were screened from the sludge samples. The bacterial colonies were inoculated into denitrification media and cultured for 48 h. The nitrate removal rate and nitrite accumulation rate were determined. Six strains with efficient nitrate degrading ability and partial denitrification performance were ultimately selected. After further purification and identification, the strain with the best partial denitrification performance and stable nitrite accumulation was obtained and was denoted as ZY04. The strain ZY04 was white, had a smooth and prominent surface, had rounded edges on the agar plate and was difficult to be selected up. The strain was identified as a Gram-negative bacterium by the Gram staining method. The DNA of strain ZY04 was extracted and amplified by PCR using the universal primers 27F and 1492R. The amplified DNA was sequenced, and the results were compared with the sequences of existing strains in GenBank. The homology between strain ZY04 and the *A. johnsonii* strain Q67 was 96%. Considering the morphology of the strain, ZY04 was identified as *A. johnsonii*. As shown in figure 1, a phylogenetic tree between strain ZY04 and other reported strains with high-efficiency denitrification was constructed using the neighbour-joining algorithm in MEGA5.0 software.

## 3.2. Amplification of functional genes

Generally, nitrate reductase (*nap/nar*) and nitrite reductase (*nirS/nirK*) are the main functional genes of the partial denitrification process. In this study, *napA* was detected and obtained for nitrate reduction, and two subunits of the *nirS* gene, *nirB* and *nirD*, were detected and obtained for nitrite reduction [38]. As shown in figures 2 and 3, the lengths of the *napA*, *nirB* and *nirD* genes were 290 bp, 206 bp and 300 bp, respectively. The *napA* gene was coupled to quinol oxidation in the periplasm via a membrane-anchored tetrahaem cytochrome to catalyse the reduction of nitrate to nitrite. The *nirS* was the $cd_1$ haem-dependent *NiR* structural gene and catalysed the translation of nitrite to NO. At the logarithmic phase of strain ZY04, the expression levels of these three genes were estimated; the *nirB* and *nirD* genes showed the same expression trends as *nirS*.

## 3.3. Modelling microbial growth kinetics

### 3.3.1. Single-substrate inhibition growth kinetics

Table 2 presents the regression results of the kinetic parameters and the regression coefficient ($R^2$). All three models, Aiba (no. 1), Edwards (no. 2) and Andrews (no. 3), could fit the growth patterns of

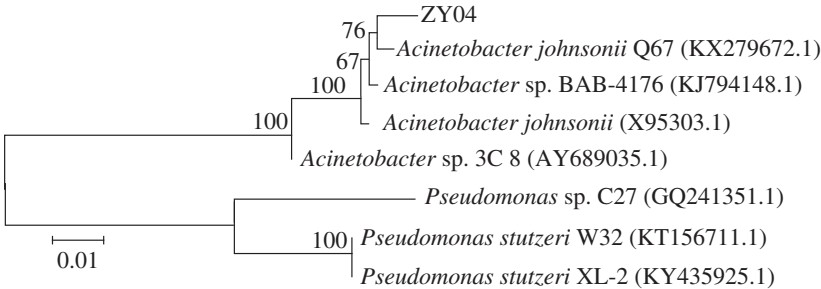

**Figure 1.** The phylogenetic tree of strain ZY04.

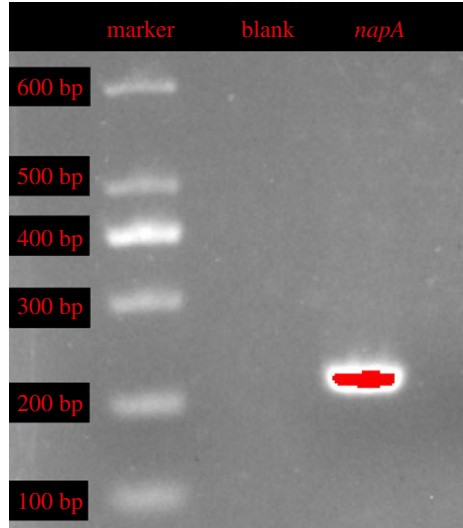

**Figure 2.** Amplification results of *napA* gene from *A. johnsonii* ZY04 (marker: 100–600 bp).

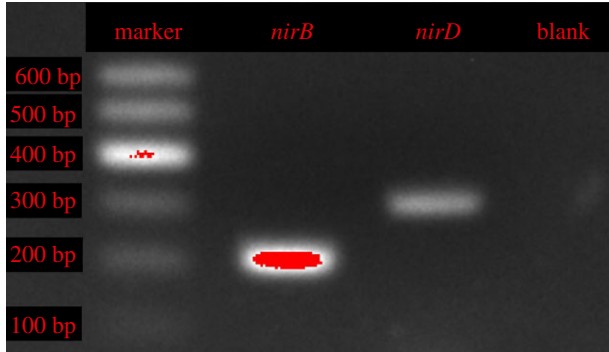

**Figure 3.** Amplification results of *nirB*, *nirD* genes from *A. johnsonii* ZY04 (marker: 100–600 bp).

strain ZY04 with $R^2$ values greater than 98% when nitrate was considered as a single inhibition substrate. The highest $R^2$ value of the Andrews model (no. 6) attained up to 99.03%. Using sodium acetate as a single inhibition substrate for variation, the performances of these three models were similar to that of nitrate as a single inhibition substrate. The $R^2$ values fitted by the three models all reached 98%, among which the $R^2$ value of the Andrews model (no. 6) was the highest and reached 99.07%. Therefore, the Andrews model was the optimal model with which to describe the growth patterns of strain ZY04 inhibited by a single substrate, $NO_3^-$-N or sodium acetate. The constants $K_S$-N and $K_i$-N of the Andrews model were calculated to be 17.14 and 1782.07, respectively, when $NO_3^-$-N was considered as the single inhibition substrate, and the constants $K_S$-C and $K_i$-C of the Andrews model were calculated to be 84.11 and 8940.23, respectively, when TOC was considered as the single inhibition substrate.

**Table 2.** Results of regression on kinetic models.

| fitting number | equation for $NO_3^-$-N | equation for TOC | Aiba $K_s$-N | $K_r$-N | $K_s$-C | $K_i$-C | Edwards $K_s$-N | $K_i$-N | $K_s$-C | $K_i$-C | Andrews $K_s$-N | $K_r$-N | $K_s$-C | $K_i$-C | $\mu_{max}$ | $R^2$ |
|---|---|---|---|---|---|---|---|---|---|---|---|---|---|---|---|---|
| single | | | | | | | | | | | | | | | | |
| 1 | Aiba | | 12.58 | 2745.61 | | | | | | | | | | | 0.5370 | 0.9874 |
| 2 | Edwards | | | | | | 29.46 | 3061.89 | | | | | | | 0.5066 | 0.9895 |
| 3 | Andrews | | | | | | | | | | 17.14 | 1782.07 | | | 0.5687 | 0.9903 |
| 4 | | Aiba | | | 62.01 | 13737.24 | | | | | | | | | 0.5367 | 0.9879 |
| 5 | | Edwards | | | | | | | 144.86 | 15317.67 | | | | | 0.5065 | 0.9897 |
| 6 | | Andrews | | | | | | | | | | | 84.11 | 8940.23 | 0.5679 | 0.9907 |
| double | | | | | | | | | | | | | | | | |
| 7 | Aiba | Aiba | 4.32 | 9377.45 | 37.69 | 19492.73 | | | | | | | | | 0.5359 | 0.9879 |
| 8 | Edwards | Aiba | | | 720.56 | 2074.63 | 0.18 | 307.99 | | | | | | | 0.0010 | — |
| 9 | Andrews | Aiba | | | 4548.64 | 7250.27 | | | | | (286.88) | $5.73 \times 10^{25}$ | | | 0.0001 | — |
| 10 | Aiba | Edwards | 228.18 | 177.22 | | | | | $1.46 \times 10^{23}$ | 3.24 | | | | | (0.0078) | — |
| 11 | Edwards | Edwards | | | | | 1.68 | 6286.85 | 147.52 | 30294.51 | | | | | 0.5055 | 0.9895 |
| 12 | Andrews | Edwards | | | | | | | 138.98 | 46102.15 | 1.81 | 3604.17 | | | 0.5171 | 0.9898 |
| 13 | Aiba | Andrews | 4.46 | 7521.37 | | | | | | | | | 43.60 | 16961.52 | 0.5461 | 0.9893 |
| 14 | Edwards | Andrews | | | | | 19.98 | 9030.79 | | | | | 43.28 | 16204.74 | 0.5348 | 0.9898 |
| 15 | Andrews | Andrews | | | | | | | | | 4.34 | 8970.72 | 48.25 | 14194.83 | 0.5523 | 0.9899 |

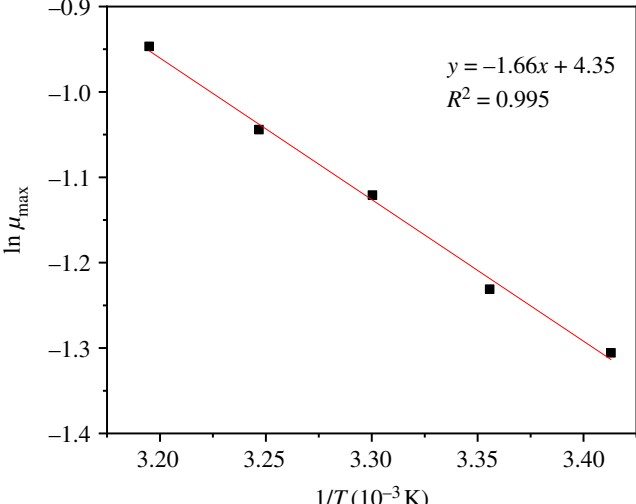

**Figure 4.** Application of the Arrhenius equation to evaluate the temperature effect on the specific microbial growth rate ($\mu$) for *A. johnsonii* ZY04.

### 3.3.2. Double-substrate inhibition growth kinetics

Nitrate and sodium acetate were used as the common substrates to establish interactive double-substrate kinetics of the growth pattern in strain ZY04, and their inhibitory effect on bacterial growth was investigated. The double-substrate models were obtained by the combination of equations (2.6)–(2.8), and the kinetic constants and regression coefficients ($R^2$) are listed in table 2. The fitting results showed that the following combinations successfully simulated the double-substrate inhibitory effects of nitrate and sodium acetate on the bacterial growth of strain ZY04 with $R^2$ values as high as 0.98: the Aiba–Aiba model (no. 7), Edwards–Edwards model (no. 11), Andrews–Edwards model (no. 12), Aiba–Andrews model (no. 13), Edwards–Andrews model (no. 14) and Andrews–Andrews model (no. 15). However, the Edwards–Aiba model (no. 8), Andrews–Aiba model (no. 9) and Aiba–Edwards model (no. 10) combinations failed to fit the experimental data; some errors occurred during the simulation process. These results demonstrate that three single-substrate models and six double-substrate models can all serve as appropriate biokinetic models for describing the bacterial growth of strain ZY04.

### 3.3.3. Kinetic model of the influence of temperature on strain ZY04 growth

As shown in figure 4, the Arrhenius equation described the growth kinetics of strain ZY04 by temperature very well; the natural logarithm of the maximum specific growth rate ($\ln\mu_{max}$) had an excellent linear relationship with the reciprocal of temperature ($1/T$). From the diagram, the pre-exponential factor ($A$) was calculated to be 77.48 days$^{-1}$, and the activation energy ($E$) was calculated to be 13.80 kJ mol$^{-1}$.

## 3.4. Denitrification characteristics and differential gene expression under different environmental conditions

### 3.4.1. Effect of C/N ratio on denitrification performance and functional gene expression

As shown in figure 5, the nitrate was rapidly degraded by strain ZY04 after a short time of adapting to the environment in all experimental groups with C/N ranging from 3 to 7. The nitrate metabolism was mainly observed at the 6–13 h stage; the removal efficiencies all reached over 90% after 13 h of degradation. Correspondingly, a remarkable accumulation of nitrite occurred during the period of 6–13 h. The highest nitrite accumulation efficiencies in the groups with C/N = 3, 5 or 7 were 80.9%, 94.2% and 87.4%, respectively. In addition, the highest nitrite accumulation efficiencies were observed in the 13th hour in the groups with C/N = 3, 6 or 7, whereas they occurred in the 16th hour in groups with C/N = 4 or 5. These results indicate that the significant accumulation of nitrite usually occurred

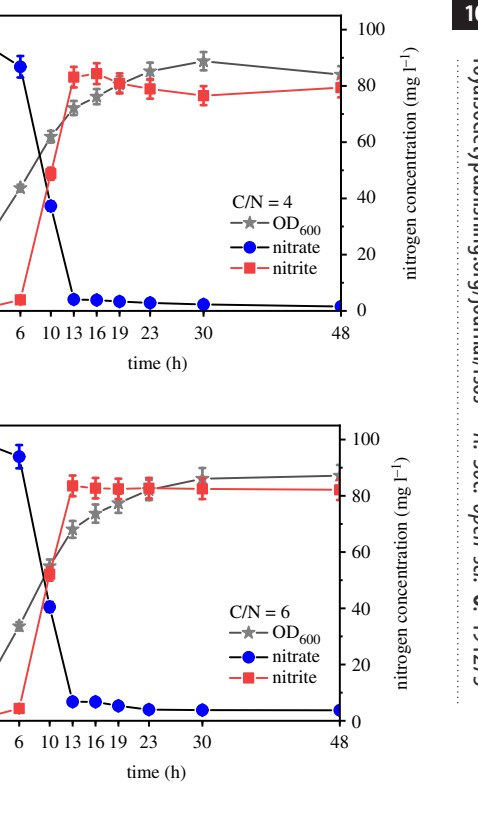

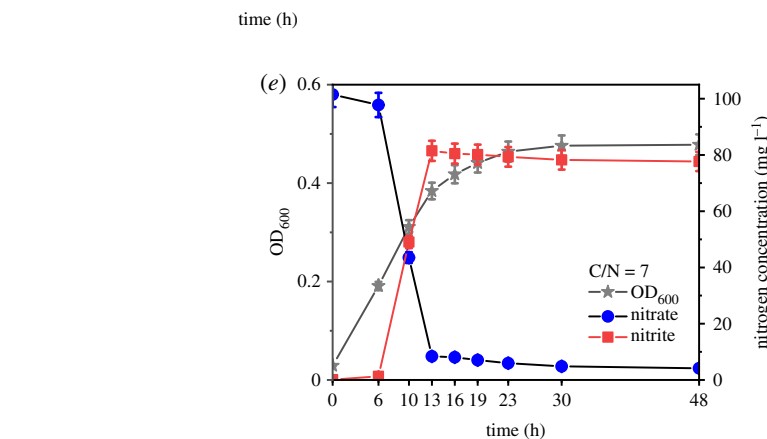

**Figure 5.** The growth and substrate concentration of strain ZY04 under different initial C/N ratio conditions.

at a later time during the cell growth phase. The nitrite accumulation efficiencies had a slight decrease after the highest values were reached, exhibiting a trend of first increasing and then decreasing in all groups. When the nitrate degradation in the experimental groups was finished, the maximum nitrite accumulation was 89.9% during the stable period in the group with C/N = 5. When the C/N ratio was 3 or 4, the nitrite accumulation decreased significantly after reaching the peak, which might have been caused by the lack of carbon source in the later stage. In this case, the nitrite removal rate was higher than the nitrate removal rate owing to the higher enzyme activity of nitrite reductase compared with nitrate reductase, thereby further leading to the decrease in nitrite accumulation. The experimental results reported by Xie *et al*. [39] also showed that the C/N ratio in denitrification had a significant influence on the nitrate reduction pathway. Ge *et al*. [40] found that a higher C/N ratio leads to an increase in nitrite accumulation and electron competition between nitrite reductase and nitrate reductase, which results in different reduction rates. The carbon source is insufficient under low C/N ratio conditions, and the nitrate reductase is less competitive with electrons than the nitrite reductase. However, with the increase in the carbon source, the competitiveness of nitrate reductase for electrons gradually increases, leading to a decline in the expression of nitrite reductase owing to the lack of carbon sources. When C/N was greater than 5, the carbon source was no longer a limiting factor for bacterial growth, and the expression levels of nitrate reductase and nitrite reductase both

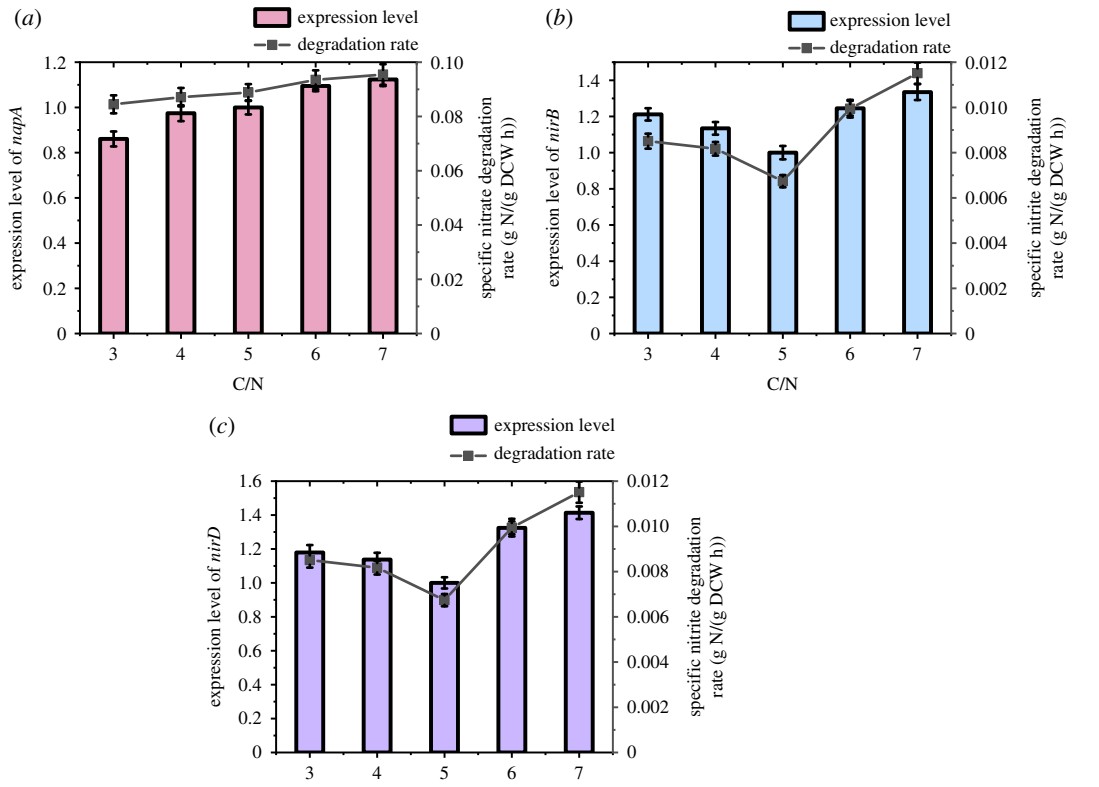

**Figure 6.** Functional genes expression and corresponding degradation rate of substrate at different initial C/N ratio conditions.

increased. With the increase in the C/N ratio, the relative expression of the *napA* gene, which encodes periplasmic nitrate reductase, increased gradually from 0.86-fold to 1.12-fold; this was consistent with the results presented in figure 6, which demonstrates that the specific nitrate reduction rate increased owing to the increase in the C/N ratio at about 10 h. There was a significant difference ($p < 0.05$) in the expression levels of the *napA*, *nirD* and *nirB* genes when the initial C/N ratio changed. With sufficient nitrate as the substrate, the expression of nitrate reductase increased with the increase in the C/N ratio, indicating that the carbon source promoted nitrate removal; this is similar to reported results in the existing literature [41]. The expression levels of the two subunits of nitrite reductase, *nirD* and *nirB*, showed trends of first decreasing and then increasing when the C/N ratio increased from 3 to 7. The expression levels of both subunits reached a minimum at C/N = 5. The relative expressions of *nirD* were 1.18-fold and 1.41-fold at C/N = 3 and C/N = 7, respectively, while the relative expressions of *nirB* were 1.21-fold and 1.33-fold at C/N = 3 and C/N = 7. Correspondingly, the specific nitrate degradation rate obtained in the batch experiment was $8.44 \times 10^{-3}$ mg l$^{-1}$ h$^{-1}$ at C/N = 3, which increased to $9.85 \times 10^{-3}$ mg l$^{-1}$ h$^{-1}$ at C/N = 4, and then decreased gradually to $6.41 \times 10^{-3}$ mg l$^{-1}$ h$^{-1}$ at C/N = 7. The results of this experiment indicate that nitrite accumulation is caused by the different expression levels of two functional genes, which encode nitrite reductase and nitrate reductase, respectively, under different C/N ratio conditions, and the difference in the denitrification rate was due to electron competition.

### 3.4.2. Effect of initial pH value on denitrification performance and functional gene expression

As shown in figure 7, there was an obvious delay period in the nitrate degradation when the initial pH value was 6 or 10, and the nitrate rapidly degraded after 10 h. The nitrate reductase might require an obvious adaptation period at pH conditions that are either too high or too low. There was no such delay of nitrate removal when the initial pH was 7–9. Additionally, no obvious lag time of cell growth was observed under the condition of pH = 7–10; however, the bacterial growth at pH = 6 was still delayed. The strain could rapidly degrade nitrate at different initial pH conditions. When the initial pH values were 6, 7, 8, 9 and 10, the degradation efficiencies reached 74%, 92%, 95%, 92% and 88% at the 16th hour, respectively. The time at which the nitrite concentration reached the maximum was

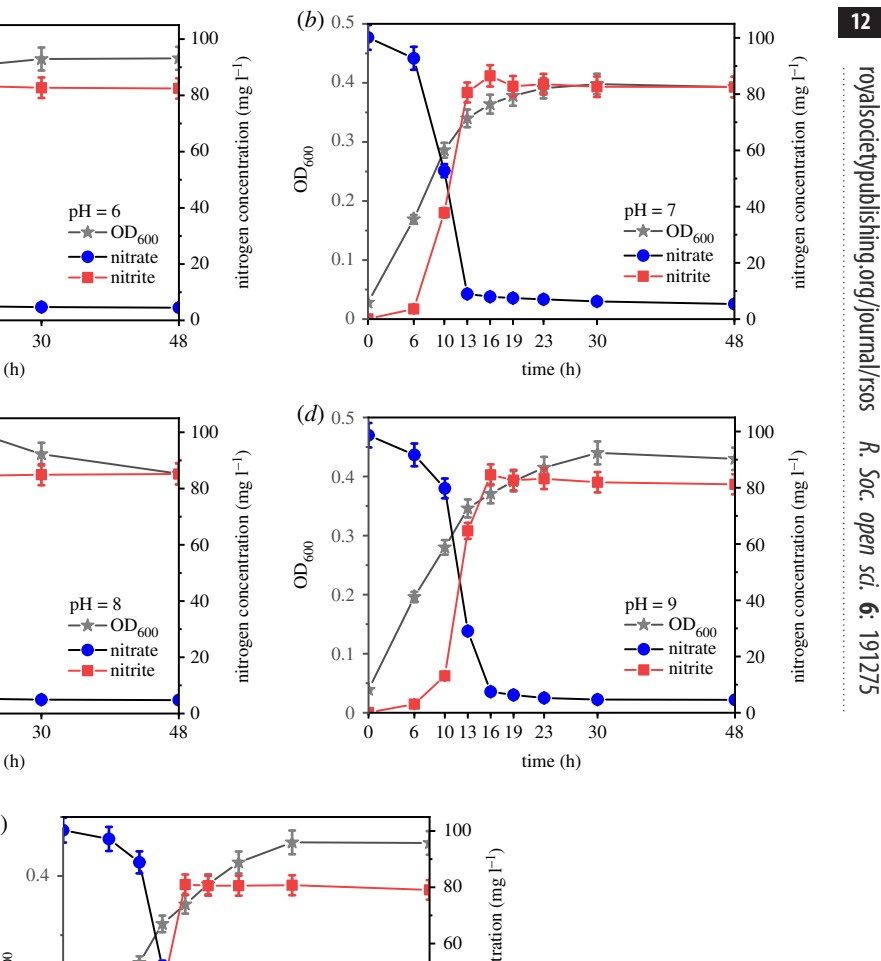

**Figure 7.** The growth and substrate concentration of strain ZY04 under different initial pH value conditions.

basically consistent with the time at which the nitrate degradation efficiency reached 90%, followed by slight degradation. The nitrite accumulation efficiencies were basically stable at approximately 80–90% at the 23rd hour and fluctuated less than 2% at the 24th hour. The nitrite accumulation efficiencies during the stable period were not significantly affected by the pH value and remained at approximately 86–88% when pH = 6–9, and slightly decreased to 83% when pH = 10. Previous works [8,25,42] have found that an increasing pH promotes the accumulation of nitrite. However, Wang *et al.* [43] demonstrated that the increase in pH value led to the decrease in the accumulation of nitrite. In the present study, the strain ZY04 could be normally grown in the pH range of 6–10. About 95% of the nitrate could be degraded within 24 h, and the nitrite accumulation efficiencies reached over 80% and remained stable for more than 24 h. These results indicate that strain ZY04 had a good tolerance to environmental pH, and the effect of pH change on nitrite accumulation was not obvious. When the C/N ratio and temperature remained constant, the effects of the initial pH on the three genes studied in this experiment were the same. The results of preliminary experimentation revealed that the strain could not grow at pH = 5 or 11; thus, the scale of the initial pH was set as 6–10 in this experiment. It is evident from the results that the initial pH value had no significant effect on the accumulation of nitrite. As shown in figure 8, there was no significant difference ($p > 0.05$) in the expression levels of the *napA*, *nirD* or *nirB* genes when the initial pH increased from 6 to 8. The change trends of the expression levels of the three genes were basically the same and exhibited a slight increase. The change of relative expressions

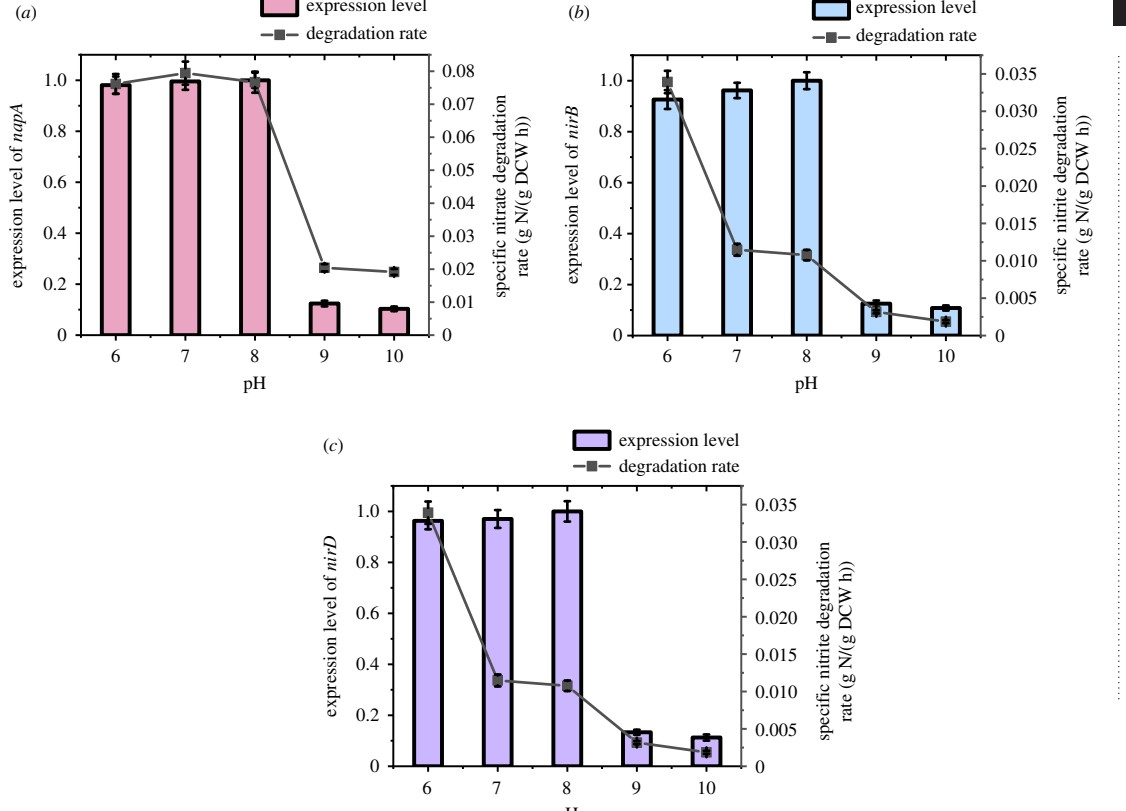

**Figure 8.** Functional genes expression and corresponding degradation rate of substrate at different initial pH value conditions.

increased by less than 0.1-fold. However, the relative expression levels of the genes dropped sharply to about 0.1-fold when the initial pH was 9–10 and decreased by more than 85% compared with that in the pH = 8 group. There was no significant difference between the relative expressions of the nitrate reductase gene and nitrite reductase gene in different initial pH environments. The suitable environment for ZY04 was an initial pH value of 6–8, the expression levels of the denitrification-related genes in the strain were dramatically inhibited at an initial pH of 9–10. A too-high initial alkalinity will suppress the denitrification process, which is accompanied by alkalinity production, by inhibiting the expression of the related functional genes.

### 3.4.3. Effect of temperature on denitrification performance and functional gene expression

The strain ZY04 was cultured under conditions of environmental temperatures ranging from 20°C to 40°C. As shown in figure 9, the strain cultivated at a low temperature of 20°C had a slow growth rate and an obvious delay period of about 9 h, whereas there was almost no delay in bacterial growth at 25°C, and a rapid increase in the $OD_{600}$ value was clearly observed after 3 h. In addition, strain ZY04 began rapid growth after 6 h of incubation at 30–40°C. It was also observed that the cell concentration with an $OD_{600}$ value of 0.20 at 40°C was not higher than that at 35°C ($OD_{600}$ = 0.28). These results indicate that temperatures that are either too high or too low could inhibit the growth of the strain and that the lag time will increase at a low temperature. The highest cell concentration, presented at 25°C, was 527.7 mg l$^{-1}$. The nitrate could be degraded by more than 90% within 12 h under the suitable growth conditions of 25–30°C, while the same removal rate occurred within 15 h at either higher or lower temperatures. The highest nitrite accumulation efficiency occurred at 25°C with a value of 89.76%, and values of 83.92%, 87.17% and 85.57% were observed at 20°C, 30°C and 40°C, respectively. The optimal temperature for nitrate degradation and nitrite accumulation was clearly 25°C. Interestingly, the nitrite accumulation rate was significantly impacted when the temperature was lower than 25°C, whereas increasing the temperature had no significant impact on the nitrite accumulation. As presented in figure 10, the relative expression of the *napA* gene first increased and then decreased when the temperature changed from 20°C to 40°C. The highest relative expression was

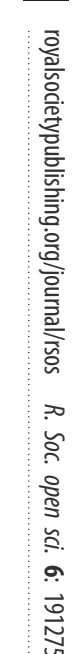

**Figure 9.** The growth and substrate concentration of strain ZY04 under different initial temperature conditions.

1.34-fold greater at 25°C than at 30°C. In addition, the transcriptional levels at 20°C and 35°C were 0.70-fold and 0.84-fold, respectively, and the lowest relative expression was 0.66-fold and was observed at 40°C. The maximum transcriptional level was more than twice as large as the minimum. There were significant differences ($p < 0.05$) in the expression levels of the *napA*, *nirD* and *nirB* genes when the culturing temperature changed. The expression trend of nitrite reductase genes at different temperatures was similar to that of nitrate reductase genes, which first increased and then decreased. Setting the experimental group at 30°C as the control, the relative expression of *nirD* increased from 0.94-fold at 20°C to the maximum value of 1.10-fold at 25°C, and then slowly decreased to the minimum of 0.74-fold at 40°C. Similarly, the relative expression of *nirB* increased from 0.85-fold at 20°C to the maximum value of 1.04-fold at 25°C, and then slowly decreased to 0.62-fold at 40°C. Unlike the nitrate reductase gene, the relative expression of the nitrite reductase gene was less variable. From 20°C to 25°C, the relative transcriptional increases of *nirD* and *nirB* were only 0.16-fold and 0.19-fold, respectively, while that of *napA* reached 0.64-fold. In addition, the relative transcriptional levels of *nirD* and *nirB* decreased by only 0.10-fold and 0.04-fold, respectively, from 25°C to 30°C, whereas that of *napA* decreased by 0.34-fold. It can be seen that both the nitrate reductase and nitrite reductase genes reached maximum gene expression at 25°C. However, the influence of temperature on the expression of these genes was different. The relative expression of the nitrate reductase gene was more sensitive to the temperature change, and the transcriptional levels of the nitrite reductase genes were relatively more stable under different temperature conditions. As shown in the nitrite accumulation study, more

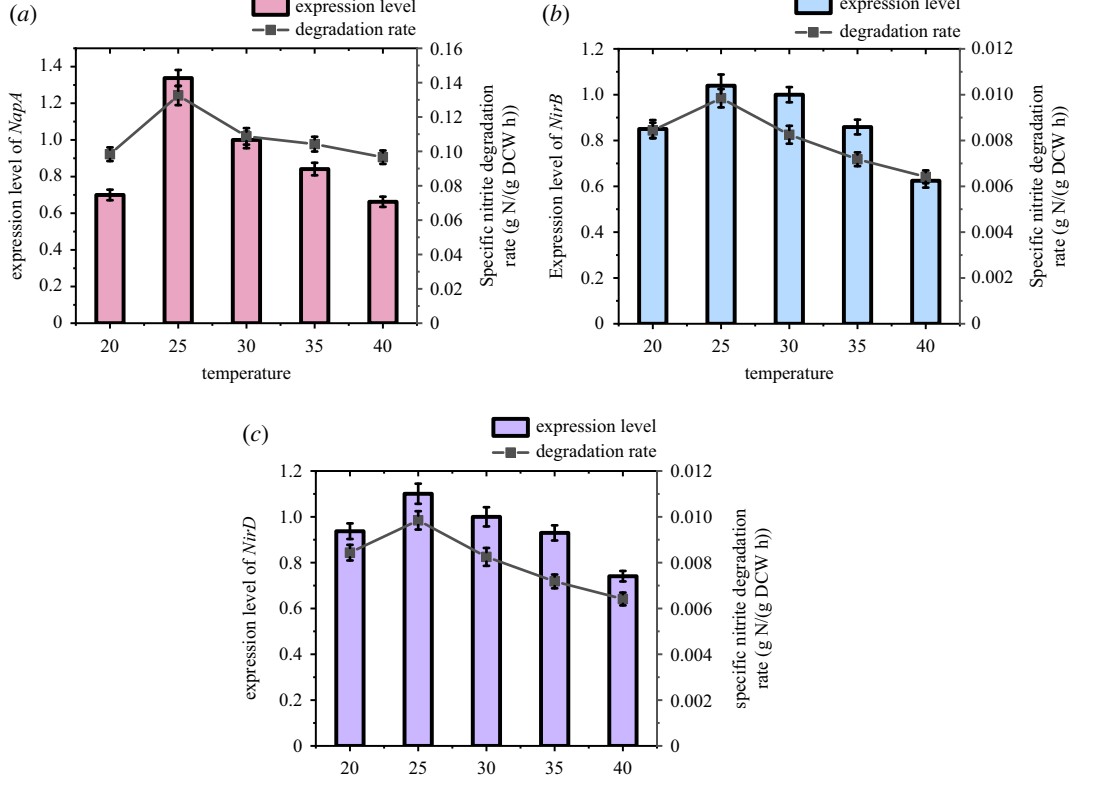

**Figure 10.** Functional genes expression and corresponding degradation rate of substrate at different initial temperature conditions.

nitrate was reduced to nitrite when the temperature was 25°C, and the amount of nitrite reduced was less than that of nitrate, leading to an efficient accumulation of nitrite.

# 4. Conclusion

In this study, a strain of *A. johnsonii* with high nitrite accumulation was successfully isolated from activated sludge, and the sequences of three key functional genes were amplified and obtained by PCR. The growth pattern of strain ZY04 was simulated by three substrate inhibitory kinetics, namely Andrews, Aiba and Edward. The results show that the Andrews model is optimal for single-substrate inhibitory growth kinetics. Five of the nine combinations of double-substrate dynamics were successfully fitted to the growth of the strain, and their regression coefficients ($R^2$) were all above 0.98. The nitrite accumulation efficiency in the strain first increased and then decreased as the C/N ratio increased from 3 to 7, and reached the maximum at C/N = 5. The relative expression of *napA* gradually increased under different C/N conditions, and the relative expressions of *nirB* and *nirD* first decreased and then increased, and reached the minimum values at C/N = 5. When the initial pH changed from 6 to 10, the nitrite accumulation rate did not change significantly. The relative expressions of *napA*, *nirB* and *nirD* increased slightly at pH 6–8, but declined by about 90% at pH 9 and 10. With the increase in temperature from 20°C to 40°C, the nitrite accumulation first increased and then decreased. The relative expression levels of *napA*, *nirB* and *nirD* also increased and then decreased, all reaching the maximum at 25°C.

Data accessibility. All DNA sequences used to construct the phylogenetic tree in the paper are stored in GenBank with accession nos. KX279672.1, KJ794148.1, X95303.1, AY689035.1, GQ241351.1, KT156711.1, KY435925.1. There are no additional experimental data.
Authors' contributions. Y.Z. and X.W. conceived and designed the experiment, Y.Z. and Z.S. performed the experiment, Y.Z. and W.W. analysed the data, Y.Z. wrote the manuscript and X.W. and J.L. revised the manuscript. All authors read and approved the final manuscript.
Competing interests. We have no competing interests.

Funding. This research was supported by the Major Science and Technology Program for Water Pollution Control and Treatment (grant no. 2017ZX07103-001).

Acknowledgements. The authors thank Sangon Biotech Co., Ltd. (Shanghai, China) for helping with the sequencing experiments carried out as part of the study.

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
