## [Reviewer comments · Royal Society Open Science]

Review History

RSOS-191275.R0 (Original submission)

Review form: Reviewer 1

Is the manuscript scientifically sound in its present form?

Yes

Are the interpretations and conclusions justified by the results?

Yes

Is the language acceptable?

Yes

Do you have any ethical concerns with this paper?

No

Have you any concerns about statistical analyses in this paper?

No

Recommendation?

Accept with minor revision (please list in comments)

Comments to the Author(s)

Thanks for the research work done by the authors, but there are some problems in the manuscript.

1. The English of your manuscript should be improved before resubmission. We suggest that you obtain assistance from a colleague who is well-versed in English or whose native language is English.
2. Please pay attention to the standardization of unit writing.

Review form: Reviewer 2

Is the manuscript scientifically sound in its present form?

Yes

Are the interpretations and conclusions justified by the results?

Yes

Is the language acceptable?

Yes

Do you have any ethical concerns with this paper?

Yes

Have you any concerns about statistical analyses in this paper?

No

Recommendation?

Accept with minor revision (please list in comments)

Comments to the Author(s)

I suggest analyzing them results using statistical tools before accepting for publication.

Review form: Reviewer 3

Is the manuscript scientifically sound in its present form?

Yes

Are the interpretations and conclusions justified by the results?

No

Is the language acceptable?

Yes

Do you have any ethical concerns with this paper?

No

Have you any concerns about statistical analyses in this paper?

Yes

Recommendation?

Major revision is needed (please make suggestions in comments)

Comments to the Author(s)

In this study, Zhang et al. isolated a new partial denitrifying bacteria, and then analyzed the degradation kinetics or the gene expression. The topic is interesting, however, the manuscript was not well presented.

- (1) In the abstract, no gene expression section was presented.
- (2) The introduction should be focused on partial denitrification rather than on anammox.
- (3) In the study, inhibition by acetate or nitrate on the biological degradation was modelled. Why acetate or nitrate would inhibit denitrification? Usually, FNA or nitrite would be a factor. Also, both inhibition would occur? The word of inhibition is suitable?
- (4) Please focus on C/N ratio further, for partial denitrification, the required organic carbon would be very small, only from +5 to +5 rather than to 0 for producing N₂. Therefore, the applied C/N ratio should be much lower than the complete denitrification.
- (5) For the pH effect, please focus on both growth and also the effect on FNA?
- (6) For temperature, the model for the temperature coefficient could be provided.
- (7) The gene expression section could be combined the system performance.
- (8) The reason for investigating of C/N, pH and temperature should be reasonably judged.

Decision letter (RSOS-191275.R0)

02-Sep-2019

Dear Miss Zhang:

Title: Investigation on growth kinetics and partial-denitrification performance in strain *Acinetobacter johnsonii* under different environmental conditions

Manuscript ID: RSOS-191275

The editor assigned to your manuscript has now received comments from reviewers. We would like you to revise your paper in accordance with the referee and Subject Editor suggestions which can be found below (not including confidential reports to the Editor). Please note this decision does not guarantee eventual acceptance.

Please submit your revised paper before 25-Sep-2019. Please note that the revision deadline will expire at 00.00am on this date. If we do not hear from you within this time then it will be assumed that the paper has been withdrawn. In exceptional circumstances, extensions may be possible if agreed with the Editorial Office in advance. We do not allow multiple rounds of

revision so we urge you to make every effort to fully address all of the comments at this stage. If deemed necessary by the Editors, your manuscript will be sent back to one or more of the original reviewers for assessment. If the original reviewers are not available we may invite new reviewers.

RSC Associate Editor:
Comments to the Author:
(There are no comments.)

RSC Subject Editor:
Comments to the Author:
(There are no comments.)

Reviewers' Comments to Author:
Reviewer: 1

Comments to the Author(s)
Thanks for the research work done by the authors, but there are some problems in the manuscript.

1. The English of your manuscript should be improved before resubmission. We suggest that you

obtain assistance from a colleague who is well-versed in English or whose native language is English.

2. Please pay attention to the standardization of unit writing.

Reviewer: 2

Comments to the Author(s)

I suggest analyzing them results using statistical tools before accepting for publication.

Reviewer: 3

Comments to the Author(s)

In this study, Zhang et al. isolated a new partial denitrifying bacteria, and then analyzed the degradation kinetics or the gene expression. The topic is interesting, however, the manuscript was not well presented.

- (1) In the abstract, no gene expression section was presented.
- (2) The introduction should be focused on partial denitrification rather than on anammox.
- (3) In the study, inhibition by acetate or nitrate on the biological degradation was modelled. Why acetate or nitrate would inhibit denitrification? Usually, FNA or nitrite would be a factor. Also, both inhibition would occur? The word of inhibition is suitable?
- (4) Please focus on C/N ratio further, for partial denitrification, the required organic carbon would be very small, only from +5 to +5 rather than to 0 for producing N₂. Therefore, the applied C/N ratio should be much lower than the complete denitrification.
- (5) For the pH effect, please focus on both growth and also the effect on FNA?
- (6) For temperature, the model for the temperature coefficient could be provided.
- (7) The gene expression section could be combined the system performance.
- (8) The reason for investigating of C/N, pH and temperature should be reasonably judged.

Author's Response to Decision Letter for (RSOS-191275.R0)

See Appendix A.

RSOS-191275.R1 (Revision)

Review form: Reviewer 3

Is the manuscript scientifically sound in its present form?

Yes

Are the interpretations and conclusions justified by the results?

Yes

Is the language acceptable?

Yes

Do you have any ethical concerns with this paper?

No

Have you any concerns about statistical analyses in this paper?

No

Recommendation?

Accept as is

Comments to the Author(s)

None

Decision letter (RSOS-191275.R1)

04-Oct-2019

Dear Miss Zhang:

Title: Investigation on growth kinetics and partial-denitrification performance in strain *Acinetobacter johnsonii* under different environmental conditions

Manuscript ID: RSOS-191275.R1

It is a pleasure to accept your manuscript in its current form for publication in Royal Society Open Science. The chemistry content of Royal Society Open Science is published in collaboration with the Royal Society of Chemistry.

RSC Associate Editor:

Comments to the Author:

The manuscript can now be accepted as the reviewer has no further comments.

RSC Subject Editor:

Comments to the Author:

(There are no comments.)

Reviewer(s)' Comments to Author:

Reviewer: 3

Comments to the Author(s)

none

Appendix A

Dear Dr. Smith,

We received your letter the other day. Thank you so much for giving us the opportunity to revise and resubmit our manuscript (RSOS-191275), entitled ‘Investigation on growth kinetics and partial-denitrification performance in strain *Acinetobacter johnsonii* under different environmental conditions’. We sincerely thank you and other three reviewers for your valuable feedback that we have used to improve the quality of our manuscript. The reviewer comments are laid out below in italicized font and specific concerns have been numbered. Our response is given in normal font and changes/additions to the manuscript are given in blue text. Besides, we have consulted a professional English editing service to help us polishing our article before submission this time. Please see the attachment for the certificate of English proofreading. We hope that the revised version of the manuscript could be considered for publication in your journal. I look forward to hearing from you soon.

With best wishes,

Yours sincerely,

Yang Zhang,
September 14, 2019

To reviewer 1:

Thank you so much for taking the time to read our manuscript and make valuable comments. We feel sorry that we do have some inadequacies in our previous manuscript. Your thoughtful comments have contributed a lot to improve the quality of our manuscript. According to your constructive suggestion, we have made some modifications to our previous draft. All changes/additions to the manuscript are given in blue text. We hope the revised manuscript could be up to your standard this time.

Comments from reviewer 1:

Thanks for the research work done by the authors, but there are some problems in the manuscript.

Specific comments:

Comment 1. *The English of your manuscript should be improved before resubmission. We suggest that you obtain assistance from a colleague who is well-versed in English or whose native language is English.*

Response: We are so appreciated that you give us the opportunity to revise and improve our article. According to your constructive suggestions, we tried our best to improve the manuscript and made some changes in the manuscript. We feel sorry for our poor writings, however, we have consulted a professional English editing service to help us polishing our article before submission this time. We hope the revised manuscript could be acceptable for you. Please see the attachment for the certificate of English proofreading.

Comment 2. *Please pay attention to the standardization of unit writing.*

Response: Thank you so much for your careful check and valuable suggestion. In view of this

suggestion, we re-checked and verified all the units that appeared in our manuscript, and revised them with blue text in the manuscript. We feel sorry for our carelessness.

To reviewer 2:

Thank you so much for giving us the opportunity to revise and resubmit our manuscript. We feel sorry that we do have some inadequacies in our previous manuscript. Your thoughtful comments have contributed a lot to improve the quality of our manuscript. According to your constructive suggestion, we do our best to make changes and additions to our previous draft. All changes/additions to the manuscript are given in blue text. We hope the revised manuscript could be up to your standard this time.

Comments from reviewer 2:

I suggest analyzing them results using statistical tools before accepting for publication.

Response: Thank you for your suggestion on using statistical tools to analyze the results. Based on this suggestion, we analyzed the data of our investigation by IBM SPSS statistics 25 and modified the manuscript according to the results. All changes/additions to the manuscript are given in blue text.

To reviewer 3:

We sincerely thank you for your professional review work on our manuscript. Your thoughtful comments and constructive suggestions have contributed a lot to improve the quality of our manuscript. As you are concerned, there are several problems that need to be addressed. According to your nice suggestions, we have made extensive corrections to our previous draft. After this revision, we have written a point-by-point response letter to you as you can see above. All changes/additions to the manuscript are given in blue text. And the detailed corrections are listed below.

Comments from reviewer 3:

In this study, Zhang et al. isolated a new partial denitrifying bacteria, and then analyzed the degradation kinetics or the gene expression. The topic is interesting, however, the manuscript was not well presented.

Specific comments:

Comment (1) *In the abstract, no gene expression section was presented.*

Response: Thank you for your advice. I'm sorry that the results of gene expression were not summarized in the abstract, which was obviously our mistake. Now we have re-discussed and supplemented this part in the abstract. You can check the supplemental part with blue text in the abstract.

Comment (2) *The introduction should be focused on partial denitrification rather than on anammox.*

Response: Thank you for your careful reminding. Partial denitrification has inseparable relationship with ANNAMOX, which was put forward to provide the nitrite substrate for ANNOMOX. In our paper we cited lots of references to describe their relationship and the

meaning of their coupling, which made the description of partial denitrification quite poor. We added more details of partial denitrification that can be seen in the introduction section in blue text.

Comment (3) *In the study, inhibition by acetate or nitrate on the biological degradation was modelled. Why acetate or nitrate would inhibit denitrification? Usually, FNA or nitrite would be a factor. Also, both inhibition would occur? The word of inhibition is suitable?*

Response: Thanks for your comments. The reasons why acetate or nitrate inhibit denitrification are as follow. First, Jacques et al.[1] found that nitrate reductase can be slowly and reversibly inactivated when exposed to high concentrations of nitrate, which means nitrate can be one of the inhibition factors of denitrification. Second, there is an upper limit of substrate concentration for bacteria, within which the bacterial growth rate will increase along with the increase of substrate concentration. However, the growth rate will decrease once the substrate concentration goes beyond the limit. This effect is often referred to as substrate inhibition, which can be due to the high osmotic pressure of a high concentration of substrate. Furthermore, it causes the cells to dehydrate and further to lead to the inhibition of growth.

Haluk Beyenal et al.[2] found that the microbial growth was limited by the concentrations of glucose and oxygen. Carbon source was generally assumed as the sole growth-limiting factor while nitrogen source was neglected in previous studies,[3] though both carbon and nitrogen were substrates for bacteria metabolism.[4] Previous works found that both ammonium and nitrite act as substrates with potential inhibitory factors simultaneously when their concentration exceeds the Anammox biomass tolerance threshold.[5 ,6] Therefore, in our study we adopted two kinds of the necessary substrate——acetate and nitrate, as the inhibition factors of the substrate inhibitory kinetics. Results of the single or double substrate inhibitory model fitting showed that both acetate and nitrate can be reasonable as the inhibiting substrate of partial denitrification.

As a substance with toxic effects on various microorganisms, FNA was not considered to be the inhibition factor in this study for the following reasons: First of all, FNA was not one of the necessary substrates of partial denitrification, so we could not consider it as the inhibition factor. If we attempt to investigate the effect of FNA on partial denitrification, the product inhibition should be more suitable. Second, combined with the experimental conditions of our study and the equation as follows, we can calculate the maximum value of FNA was 0.0189 mg/L.

$$FNA = \frac{47}{14} \times \frac{\rho[\text{NO}_2^- - \text{N}]}{e^{\left(\frac{-2300}{273 + T}\right)} \times 10^{\text{pH}}}$$

FNA is determined by the level of nitrite accumulated and pH in biological nitrogen removal system. We can observe that the maximum concentration of FNA was 0.452mg/L at 16th when pH value was 6.0 through our experiment data. However, the average nitrate degradation rate was still remaining at a high level of 7.13 mg/(L•h) between 16 h to 19 h, which indicated that the nitrate degradation ability of ZY04 had not been affected significantly. Whereas the concentration of FNA in substrate inhibition experiment was far lower than 0.452 mg/L, we hold the believe that

the FNA inhibition on partial denitrification in our paper could be ignored. Due to the short research period of partial denitrification, there was little published literature about the partial denitrification product——nitrite leading to FNA inhibiting partial denitrification. Therefore, we consulted the studies of FNA inhibition on denitrifying phosphorus-accumulating organisms in our answer. The nitrate reduction activity of the biomass was observed to be totally inhibited when FNA level was greater than the threshold concentration (0.2 mg HNO₂⁻-N/L).[7] The inhibitory effect of denitrification by PAOs is considerably weaker than that on phosphorus uptake.[8, 9]

Reference

- 1 Jacques, J. G. J., Burlat, B., Arnoux, P., Sabaty, M., Guigliarelli, B., Leger, C., Pignol, D., Fourmond, V. 2014 Kinetics of substrate inhibition of periplasmic nitrate reductase. *Bba-Bioenergetics*. 1837, 1801-1809. (10.1016/j.bbabi.2014.05.357)
- 2 Beyenal, H., Chen, S. N., Lewandowski, Z. 2003 The double substrate growth kinetics of *Pseudomonas aeruginosa*. *Enzyme and Microbial Technology*. 32, 92-98. (Pii S0141-0229(02)00246-6
Doi 10.1016/S0141-0229(02)00246-6)
- 3 Emerald, F. M. E., Prasad, D. S. A., Ravindra, M. R., Pushpadass, H. A. 2012 Performance and biomass kinetics of activated sludge system treating dairy wastewater. *Int J Dairy Technol*. 65, 609-615. (10.1111/j.1471-0307.2012.00850.x)
- 4 Chen, J., Zhao, B., An, Q., Wang, X., Zhang, Y. X. 2016 Kinetic characteristics and modelling of growth and substrate removal by *Alcaligenes faecalis* strain NR. *Bioproc Biosyst Eng*. 39, 593-601. (10.1007/s00449-016-1541-9)
- 5 Strous, M., Kuenen, J. G., Jetten, M. S. M. 1999 Key physiology of anaerobic ammonium oxidation. *Appl Environ Microb*. 65, 3248-3250.
- 6 Dapena-Mora, A., Fernandez, I., Campos, J. L., Mosquera-Corral, A., Mendez, R., Jetten, M. S. M. 2007 Evaluation of activity and inhibition effects on Anammox process by batch tests based on the nitrogen gas production. *Enzyme and Microbial Technology*. 40, 859-865. (10.1016/j.enzmictec.2006.06.018)
- 7 Ma, J., Yang, Q., Wang, S. Y., Wang, L., Takigawa, A., Peng, Y. Z. 2010 Effect of free nitrous acid as inhibitors on nitrate reduction by a biological nutrient removal sludge. *Journal of Hazardous Materials*. 175, 518-523. (10.1016/j.jhazmat.2009.10.036)
- 8 Zhou, Y., Pijuan, M., Yuan, Z. G. 2007 Free nitrous acid inhibition on anoxic phosphorus uptake and denitrification by poly-phosphate accumulating organisms. *Biotechnol Bioeng*. 98, 903-912. (10.1002/bit.21458)
- 9 Zhou, Y., Ganda, L., Lim, M., Yuan, Z. G., Kjelleberg, S., Ng, W. J. 2010 Free nitrous acid (FNA) inhibition on denitrifying poly-phosphate accumulating organisms (DPAOs). *Appl Microbiol Biot*. 88, 359-369. (10.1007/s00253-010-2780-3)

Comment (4) *Please focus on C/N ratio further, for partial denitrification, the required organic carbon would be very small, only from +5 to +5 rather than to 0 for producing N₂. Therefore, the applied C/N ratio should be much lower than the complete denitrification.*

Response: Thank you for your valuable suggestions. Partial nitrification processes could be a valuable tool, as theoretically 1.71 mg and 2.86 mg COD is consumed for degrading per mg NO₂⁻-N and NO₃⁻-N in denitrification,[1] respectively. Partial nitrification reduces the aeration

consumption by 25%, while the subsequent denitrification reduces the need for organic matter by 40%.[2] However, partial denitrification could be impossible according to the theoretical C/N ratio in practical applications. For examples, Zhang et al.[3] investigated low C/N (3.19 in average) sewage with the chemical oxygen demand (COD) concentration of 180.14 mg/L. The work of Ji et al.[4] showed a stable, high nitrogen removal efficiency (90%) with an effluent total nitrogen of 5.8 mg N/L under low C/N (~2.9). It can be observed that the C/N ratio required can be very different due to the different research conditions when partial denitrification is applied to practical research. It is because of the difference in electronic competition between nitrate reductase and nitrite reductase under different environmental conditions. Partial denitrification needs to be achieved under the conditions where the nitrate reductase activity is optimal and the nitrite reductase activity is poor. In our study, we stood on pure bacteria which was different from other reported studies, and the C/N ratio conditions set in this experiment are determined according to the performance in the preliminary experiment.

Reference

- 1 Jenni, S., Vlaeminck, S. E., Morgenroth, E., Udert, K. M. 2014 Successful application of nitrification/anammox to wastewater with elevated organic carbon to ammonia ratios. *Water Research*. 49, 316-326. (10.1016/j.watres.2013.10.073)
- 2 Turk, O., Mavinic, D. S. 1989 Maintaining Nitrite Buildup in a System Acclimated To Free Ammonia. *Water Research*. 23, 1383-1388. (Doi 10.1016/0043-1354(89)90077-8)
- 3 Zhang, T., Wang, B., Li, X. Y., Zhang, Q., Wu, L., He, Y., Peng, Y. Z. 2018 Achieving partial nitrification in a continuous post- denitrification reactor treating low C/N sewage. *Chemical Engineering Journal*. 335, 330-337. (10.1016/j.cej.2017.09.188)
- 4 Ji, J. T., Peng, Y. Z., Mai, W. K., He, J. Z., Wang, B., Li, X. Y., Zhang, Q. 2018 Achieving advanced nitrogen removal from low C/N wastewater by combining endogenous partial denitrification with anammox in mainstream treatment. *Bioresource Technol*. 270, 570-579. (10.1016/j.biortech.2018.08.124)

Comment (5) *For the pH effect, please focus on both growth and also the effect on FNA?*

Response: Thank you for your thoughtful advice. We feel sorry that our description about the effect of pH value on the microbial growth was poor, therefore we added more explanation in section 4.2.2 in blue text. As for the effect of FNA, it is well known that a decrease in pH value leads to an increase of FNA concentration under the same conditions. We can observe in Fig. 1 that the maximum concentration of FNA was 0.452mg/L at 16th when pH value was 6.0. Between 16 h and 19 h, the nitrate degradation rate was 7.13 mg/(L•h), which still remained at a high level. Besides, the nitrate degradation rate was not decreased until the nitrate concentration was under 5 mg/L. In our study, the concentration of FNA was increased gradually with the degradation of nitrate and accumulation of nitrite. At the same time, because the whole process came up in only 23 h, we considered the main reason of the nitrate degradation rate significantly reducing to be the lack of substrate rather than inhibition effect of FNA.

Comment (6) For temperature, the model for the temperature coefficient could be provided.

Response: Thank you for your constructive suggestions which would make our manuscript more plentiful. According to your suggestion, we used the modified Arrhenius equation to fit the experimental data of the effect of temperature on microbial growth. The results showed that the equation could well explain the influence of temperature on the growth of strain ZY04. The changes were made in section 3.5 and 4.3.3 with blue text.

Comment (7) The gene expression section could be combined the system performance.

Response: Thanks for your valuable comments about our manuscript. We feel sorry that our description of genes expression did not combined with the system performance in the paper. Based on your suggestion, we have combined the parts of ‘Denitrification characteristics under different environmental conditions’ and ‘Functional genes amplification and differential gene expression at different environments’ to be a new section ‘4.4 Denitrification characteristics and gene expression under different environmental conditions’. We also supplemented the relation between gene expression and system performance to make it easier to understand. All changes/additions to the manuscript are given in blue text.

Comment (8) The reason for investigating of C/N, pH and temperature should be reasonably judged.

Response: Thank you very much for pointing out this problem, we feel sorry that we did not provide enough information about the reasons for our investigation. In the last paragraph of the introduction section, the purpose and contents of our research have been supplemented in blue

text. We hope the revised manuscript could be up to your standard this time.